# Semantic knowledge graph fusion for fake news detection: Unifying content-based features and evidence-based analysis in the COVID-19 infodemic

Rayees Ahmad Dar[1]*, Rana Hashmy[1], Muhammad Shahid Anwar[2], Patrik Böhm[3], Jaroslav Frnda[3]

1 Department of Computer Science, University of kashmir, Srinagar, J&K, India, 2 Department of AI and Software, Gachon University, Seongnam-si, South Korea, 3 Department of Quantitative Methods and Economic Informatics, Faculty of Operation and Economics of Transport and Communications, University of Zilina, Zilina, Slovakia

* rayees.csscholar@kashmiruniversity.net

**Data availability statement:** The datasets and ontologies utilized in this study are publicly

## Abstract

In the era of digital communication, the rapid spread of information has brought both benefits and challenges. While it has democratized access to knowledge, it has also led to an increase in fake news, with significant societal repercussions. The COVID-19 pandemic has exacerbated this issue, resulting in what the World Health Organization has termed an "infodemic." In light of this, developing effective methods for detecting fake news is of paramount importance. In this paper, we introduce a novel approach that integrates knowledge graphs and Named Entity Recognition (NER) based on a biomedical language model to address the challenge of fake news detection. Our method aims to enhance detection accuracy by combining content analysis with entity-level insights. Our approach involves three key components. First, content analysis uses a contextual language model to capture the semantic context of the content, enabling the extraction of meaningful insights essential for identifying fake news. Second, the NER component, built on a biomedical language model, precisely identifies and categorizes entities within the content, offering a deeper understanding crucial for detecting misinformation in the biomedical domain. Finally, entity integration employs knowledge graph embeddings to transform identified entities into a format that facilitates enhanced processing and detection. By blending these components, our method creates a unified representation of the content, incorporating both semantic context and entity-based insights. This comprehensive approach significantly improves the accuracy of fake news detection. Our extensive experiments demonstrate the effectiveness of this method, particularly in the early identification of false information. The results underscore the potential of our approach as a powerful tool in combating misinformation, particularly in critical areas such as public health.

available, and have been deposited in a Zenodo repository along with the code for implementation. They are accessible at https://doi.org/10.5281/zenodo.12669980.

**Funding:** The author(s) received no specific funding for this work.

**Competing interests:** The authors have declared that no competing interests exist.

## Introduction

In the ever-evolving landscape of information dissemination, the insidious spread of fake news has become a pervasive concern, particularly on influential social media platforms such as Twitter [1–3]. The transformative impact of these platforms in shaping public opinion amplifies the urgency of addressing the rapid proliferation of false information [4,5]. Beyond the immediate threat of misinformation, fake news engenders far-reaching consequences, including financial, social, and emotional risks to society at large [6,7]. Consider, for instance, the financial implications of fake news influencing stock markets. In recent years, incidents where misinformation on social media led to sudden stock fluctuations have underscored the tangible economic consequences [8]. Furthermore, the societal impact of politically motivated fake news, capable of inciting unrest or influencing election outcomes, is a critical concern. In addition to the financial and societal impacts, the dissemination of fake news undermines public trust in institutions and erodes social cohesion [9]. During times of crisis, such as public health emergencies or political upheavals, misinformation can sow confusion and division among communities. When individuals are exposed to false or misleading information, it distorts their understanding of reality and fosters skepticism toward authoritative sources of information. Consequently, public trust in institutions such as governments, media organizations, and scientific establishments is undermined, impeding efforts to address pressing issues and hindering societal progress [10]. Therefore, combating fake news is essential for preserving the integrity of information and fostering trust and solidarity within society. Navigating the intricate landscape of fake news detection demands innovative methodologies, especially in the face of increasingly sophisticated misinformation techniques. Traditional approaches, often centered around language analysis or predefined patterns, prove inadequate against the intricacies of well-crafted manipulation strategies. For instance, analyzing linguistic patterns alone might fall short in identifying subtly crafted fake news pieces designed to mimic legitimate reporting [11]. Hence, there is a critical need for innovative and versatile detection frameworks capable of adapting to evolving patterns of deception. By leveraging advanced technologies such as natural language processing, machine learning, and knowledge representation, researchers can develop sophisticated algorithms that identify fake news and anticipate emerging trends in misinformation dissemination. Moreover, the incorporation of domain-specific knowledge is paramount. Consider the domain of health-related misinformation during a pandemic, where erroneous information about preventive measures, treatments, or vaccine efficacy can have dire consequences. In response, this study advocates for a comprehensive strategy that integrates Knowledge Graphs (KG) and harnesses—BioBERT [12]—based Named Entity Recognition (NER) in tandem with linguistic analysis to enhance detection accuracy. Drawing inspiration from recent advancements, such as the utilization of KGs in enhancing contextual understanding, we suggest that incorporating background knowledge is pivotal in the fight against misinformation [13]. To illustrate the multifaceted nature of misinformation, consider the scenario of health-related fake news. During a public health crisis, misinformation regarding treatment methods or vaccine efficacy can lead to real-world consequences, such as individuals opting for unverified treatments or refusing vaccination.

The proposed methodology, encapsulated in the CogiGraph framework, comprises three fundamental components: (1) Content Encoding, (2) Named Entity Recognition, and (3) Knowledge Graph Integration. These components synergistically contribute to a dual objective—capturing nuanced meanings within news content and comprehending intricate relationships among entities, empowering accurate decisions in fake news detection. Delving into the Content-Encoding component, we employ DistilBERT [14] to uncover

semantic nuances embedded in news articles. This approach allows us to analyze the contextual subtleties often exploited in fake news to mimic genuine reporting [15]. The Named Entity Recognition (NER) component utilizes a BioBERT-based model, allowing for precisely identifying entities within the text. This is crucial in detecting misinformation that might manipulate the representation of entities or events [16]. Finally, the third component utilizes simplE [17] for graph embeddings, facilitating the seamless integration of Knowledge Graphs to enhance the understanding of relationships among entities and topics. This becomes particularly relevant in scenarios where fake news leverages complex networks of entities to propagate misinformation, as seen in politically motivated campaigns [18]. To evaluate the efficacy of the CogiGraph framework, comprehensive analyses are conducted on the "Constraint@AAAI 2021 COVID-19" dataset [19]. The results showcase the superior performance of our framework compared to state-of-the-art methods. In an era where the urgency for early detection of fake news intensifies, this paper makes a significant contribution by introducing a comprehensive framework. By seamlessly combining Knowledge Graphs and BioBERT-based Named Entity Recognition (NER) with detailed content analysis, our approach focuses solely on textual content to enable early and accurate detection of fake news.

In conclusion, as the rapid dissemination of misinformation challenges the credibility of information, this research endeavors to lay the groundwork for fostering credible and factual information. By integrating domain-specific knowledge graphs curated from reputable ontology and datasets, our detection framework gains the ability to discern the validity of health-related claims with higher accuracy. Additionally, utilizing specialized named entity recognition models fine-tuned on domain-specific corpora further enhances the precision of entity identification within health-related news articles. Therefore, our approach not only addresses the broader challenge of fake news detection but also offers tailored solutions to combat misinformation within specific domains, thereby safeguarding public health and well-being to navigate the evolving information landscape contributing to the ongoing dialogue on fortifying the integrity of our digital discourse.

## Related work

Researchers have proposed various detection methods to address the challenge of fake news. This section reviews existing literature that analyzes content, employs knowledge-based methods, and integrates contextual information using knowledge graphs.

### Content-based approaches

Content analysis methods, focusing on the language and structure of news, have gained prominence in fake news detection [20]. Some studies utilized conventional machine learning algorithms for misinformation detection. For instance, hybrid features and decision trees achieved high accuracy in Twitter propaganda detection [21], highlighting the potential of content-based features. Advanced deep learning techniques, such as attention mechanisms, delve into the intricate details of language and context [22]. For example, a study investigates the potential of Long Short-Term Memory (LSTM) networks for content-based spam detection on social media. Another hybrid deep learning model, combining convolutional and recurrent neural networks (RNNs), demonstrated superior performance in fake news classification [23].

Recent research has explored the impact of various embedding techniques on fake news detection. The use of word embeddings has been shown to enhance detection accuracy by capturing semantic relationships between words [24]. Transformer-based embeddings, such as those derived from models like BERT, have also been applied to fake news detection tasks,

leveraging their ability to model complex language patterns [25]. Furthermore, document embeddings, which represent entire texts as vectors, have been utilized to improve detection performance by encapsulating contextual information [26].

In addition to embedding techniques, real-time architectures have been proposed to address the need for timely fake news detection. These systems are designed to process and analyze news content rapidly, enabling prompt identification of misinformation [27]. Moreover, incorporating social network features, such as user interactions and propagation patterns, has been found to enhance detection models by providing additional context beyond the content itself [28].

The challenge of detecting fake news in multilingual contexts has also been addressed through the use of multilingual transformers. These models are capable of understanding and processing multiple languages, making them suitable for detecting misinformation across diverse linguistic landscapes [29].

While these approaches show promise, concerns about bias and limited use of external knowledge persist. Bias from potentially biased training data and fact-tempering attacks can lead to inaccurate predictions and hinder generalizability [30,31]. Potthast et al. [32] identified false information by analyzing the distinct characteristics of textual material using a meta-learning methodology. Deep neural networks have also been used to acquire detection features, circumventing the expensive process of manually designing features. Kong et al. [33] used a combination of bidirectional long short-term memory (Bi-LSTM) and a convolutional neural network (CNN) to identify bogus news accurately. This approach was chosen due to the capacity of these models to represent textual material efficiently. Zhao et al. [34] used a mixture-of-experts model to integrate detection characteristics from many domains, improving performance.

## Knowledge-based approaches

Knowledge-based approaches use external information sources and knowledge databases to verify the factual accuracy of claims. Early research employed web-based statistical analysis for fact-checking [35]. Computational fact-checking introduced automatic extraction and verification of claims against structured knowledge databases [36]. Knowledge graphs have been leveraged to navigate relationships between entities and verify factual accuracy [37]. Hybrid systems analyzing linguistic features and external knowledge sources have shown improved fake news detection performance [38]. While knowledge-based approaches demonstrate effectiveness, challenges remain, including ensuring data reliability, addressing limited knowledge base coverage, and adapting to the evolving nature of fake news. Future research directions involve developing robust methods for assessing data reliability, expanding knowledge base coverage, and investigating adaptable machine learning models [39]. Zhang et al. [40] introduced a multimodal knowledge-aware event memory network to identify rumors. More precisely, a knowledge-aware network was built to incorporate external information from real-world knowledge graphs as supplementary evidence. Furthermore, they developed an event memory network to acquire event-invariant characteristics as a benchmark for achieving more resilient representations. A system was developed in [41] that used a knowledge graph to identify and explain bogus news. The retrieved graph embeddings were merged with a graph convolutional network to get the detection outcomes. Li et al. [42] used factual information and subjective opinions to identify false news by creating diverse graph structures. While including external knowledge into the detection process might enhance the reliability of findings by examining the links between items in the knowledge graph, the specific approach to combining text information with external knowledge remains unresolved.

## Network immunization for fake news detection

Network immunization techniques aim to prevent the spread of misinformation in social networks by identifying and mitigating fake news at the source. These methods leverage graph-based strategies to identify influential nodes and control the dissemination of malinformation.

Community detection techniques, for instance, have been used to isolate clusters of users involved in spreading fake news. These approaches analyze network structures to identify communities and immunize them effectively, thereby curbing the propagation of misinformation [43].

Weighted directed spanning trees offer another effective solution for mitigating fake news in real time. By organizing network nodes into hierarchical structures, these trees enable quick identification of fake news sources and facilitate timely intervention [44].

Budget-based immunization algorithms focus on optimizing resource allocation to achieve maximum impact in preventing the spread of fake news. These algorithms prioritize key nodes for immunization based on their influence in the network, ensuring efficient use of limited resources [45].

These strategies demonstrate the potential of network-based methods in complementing content-based and knowledge graph approaches, offering a holistic solution to the fake news detection problem.

## Knowledge graphs and fake news detection

Modern approaches leverage graph structures for fake news detection. A Graph-based Markov Chain approach segregates real and fake news articles, utilizing random walks to assess similarity [46]. Knowledge graphs (KGs), interconnected databases containing entities and relationships, have been used in recent studies [47]. Models like TransE [48] and DistMult [49] have been proposed for embedding entities and relationships within KGs. Researchers have explored using KGs for link prediction within the context of fake news detection. KGs have been employed for content-based fake news detection, demonstrating the effectiveness of external knowledge sources [50]. Graph neural networks (GNNs) alongside KGs have been proposed, achieving promising results in fake news detection tasks [51,52]. To identify false news, Vaibhav et al. [53] used document sentences as graph nodes to represent documents as graph structures. They next employed graph attention networks to acquire document characteristics. In addition, news shared on social media platforms may include many types of information, such as text, user, and temporal data, that may be used for detecting purposes. Nguyen et al. [54] introduced a technique for learning graphical representations that accurately capture the social context of false news. Zhang et al. [55] developed a heterogeneous network that incorporates news items, authorship, and news topics. They established a deep, diffusive network to combine this information to detect false news. Furthermore, false information often disseminates rapidly via social media platforms, text messaging, or electronic mail. Hence, by using deep learning methods, the identification of false news may be achieved by examining the velocity and extent of news transmission. Additional studies, such as [56], have suggested using graph convolutional networks to represent the spread of news. Dou et al. [57] established a false news detection system that considers user preferences and integrates the spread of news and associated topics. Despite advancements, challenges remain in effectively considering the complex relationships within news articles and interconnected KG information. Future research involves developing robust methods for integrating diverse types of knowledge from KGs, exploring advanced machine learning models, and designing efficient algorithms for large-scale KGs and real-time fake news detection

systems. Our approach builds upon these foundations by integrating content-based analysis, knowledge-based validation, and KG-driven enrichment. By blending entities and relationships from KGs with content-based representations, our method aims for a comprehensive understanding of news articles, enhancing the effectiveness of fake news detection, particularly in early detection scenarios.

## Methodology

In this section of our research paper, the authors explain the approach we've developed. We outline our steps, demonstrating how we've combined different techniques to create a solid foundation for our system.

We first discuss the semantic encoding of the news content, the KG construction and enrichment, the graph embedding extraction, and finally, the fusion and prediction sections.

### Text preprocessing

We have used tweet-preprocessor (pypi.org/project/tweet-preprocessor/) library to filter out unnecessary data, such as URLs, emoticons, username handles, etc., from the tweets.

### Semantic encoding through DistilBERT

Our methodology utilizes DistilBERT as the primary tool for extracting contextual information from the text. DistilBERT is a more resource-efficient variant of BERT, designed to encode the core content of the text into contextual embeddings. These embeddings serve as mathematical representations of the essential information contained within the text, facilitating further analysis. The process involves tokenizing the input text using DistilBERT's tokenizer, which breaks the text into a sequence of tokens. This token sequence is then passed through the DistilBERT model, which is composed of stacked bidirectional transformer encoders. These layers enable DistilBERT to capture contextual and bidirectional relationships between words, resulting in rich semantic embeddings. Mathematically, the process can be represented as follows:

$$E_{\text{context}}(X) = \text{DistilBERT}(X) \tag{1}$$

where

- $X$ represents the input text.
- $E_{\text{context}}(X)$ denotes the contextual embeddings generated by DistilBERT.

This approach allows us to efficiently capture the contextual information within the text, providing a solid foundation for subsequent analysis and classification tasks in our fake news detection system.

### Constructing a knowledge graph

Our approach to constructing a Knowledge Graph (KG) for COVID-19 knowledge begins with the selection of the COVID-19 ontology [58] as the foundational framework. This ontology, developed through collaborative efforts across multiple disciplines, is a structured system for organizing essential entities and concepts relevant to COVID-19 research. Formally, the COVID-19 ontology is denoted as $\mathcal{O}_{COVID-19}$.

**Initial approach:** The construction of the KG follows these steps:

1. **COVID-19 ontology selection**: Choose the COVID-19 ontology $\mathcal{O}_{COVID-19}$ as the basis for KG construction.
2. **Entity identification and mapping**: Identify entities and concepts from $\mathcal{O}_{COVID-19}$ and map them to nodes in the KG schema.
3. **Relationship establishment**: Define relationships within $\mathcal{O}_{COVID-19}$ and map them to edges in the KG.
4. **Node and edge creation**: Create nodes and edges in the KG based on the identified entities, concepts, and relationships.

**Refinement and enrichment:** Further refinement and enrichment of the KG involve:

- **Data integration**: Incorporate additional information from scientific literature, public health reports, and other relevant sources to enrich the KG.
- **Semantic enhancement**: Refine entity types, relationships, and attributes to improve semantic representation within the KG.
- **Evaluation and feedback**: Iteratively refine the KG based on evaluation metrics and feedback from domain experts to ensure alignment with fundamental COVID-19 facts.

The construction process is formalized as follows:

$$\mathcal{G} = \text{ConstructKG}(\mathcal{O}_{COVID-19})$$

where $\text{ConstructKG}(\cdot)$ denotes the KG construction algorithm utilizing the COVID-19 ontology $\mathcal{O}_{COVID-19}$. Through this iterative construction, refinement, and enrichment process, we aim to develop a comprehensive and accurate representation of COVID-19 knowledge within the KG framework.

## KG enrichment

To enhance the knowledge encoded in our Knowledge Graph (KG), we leveraged the CORD-19 dataset [59], which contains a vast collection of scholarly articles related to COVID-19. We focused on the abstracts and introductions of a subset of papers from BioRxiv and PMC. We employed advanced natural language processing (NLP) techniques for entity extraction and relationship inference. We utilized the BioBERT model [12], a domain-specific language model pre-trained on biomedical text for entity extraction. To adapt BioBERT for our specific task of entity recognition, we fine-tuned it using a processed version of the CORD-19 dataset Processed Version of CORD-19 Dataset for NER: https://www.kaggle.com/datasets/sushilkumarinfo/cord19processeddataset. Entity recognition involves assigning entity labels $Y = (y_1, y_2, ..., y_n)$ to input tokens $X$, achieved through token-level classification facilitated by a linear transformation $W_{\text{NER}}$ and softmax activation, as depicted in Equation 2.

$$P(y_i|x_i) = \text{Softmax}(W_{\text{NER}} \cdot x_i) \tag{2}$$

Next, we extracted relationships embedded within textual knowledge using the Stanford OpenIE NLP tool [60], in conjunction with the named entities recognized by the fine-tuned BioBERT model. This process yielded triplets $(s,p,o)$ representing complex relationships within text.

$$\text{Triples} = \text{OpenIE}(X) \tag{3}$$

We also exclusively incorporate factual news from the training data to uphold the KG's integrity and reliability. This deliberate selection ensures the KG's utility and credibility, particularly when verifying news items. Conversely, fake news from the training dataset is deliberately omitted to maintain the accuracy and trustworthiness of the KG, safeguarding its quality for subsequent analyses. The resulting enriched KG, denoted as $\mathcal{G}_{\text{enriched}}$, provides a comprehensive depiction of COVID-19 entities and their intricate relationships. A high-level view of our Knowledge Graph is illustrated in Fig 1.

### Alignment scores and KG embeddings

Alignment scores and KG embeddings play a pivotal role in our fact verification approach, facilitating the assessment of alignment between news items and factual knowledge stored within the knowledge graph. This evaluation offers insights into the degree of similarity between a news item and the existing knowledge, effectively quantifying its truthfulness concerning the KG. Consider a news item represented by a set of named entities $\mathcal{M} = \{m_1, m_2, ..., m_k\}$ and relationships $\mathcal{R} = \{r_1, r_2, ..., r_l\}$, extracted using the finetuned BioBERT model and the OpenIE tool, as detailed in the previous subsection. Alignment scores are computed to gauge the similarity strength of each entity $m_i$ compared to entities in the knowledge graph.

The alignment score $\text{score}(m_i)$ for each entity mention $m_i$ and its corresponding KG mention is calculated as follows:

$$\text{score}(m_i) = \frac{\text{similarity}(m_i, \text{kg\_mention})}{\text{max\_length}(m_i, \text{kg\_mention})} \tag{4}$$

Here, $\text{similarity}(m_i, \text{kg\_mention})$ represents the similarity between the news item mention and the corresponding KG mention, while $\text{max\_length}(m_i, \text{kg\_mention})$ denotes the

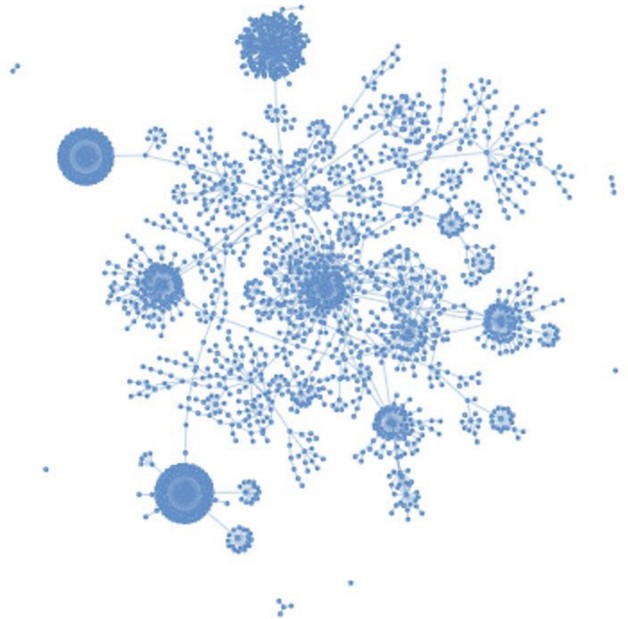

**Fig 1. High-level view of the knowledge graph.**

maximum possible length of either mention. We employ a similarity calculation procedure to quantify each entity's similarity and the relationship between the news items and those in the knowledge graph. This involves obtaining textual representations of the named entities extracted from the news items and the knowledge graph. Subsequently, we compute word embeddings for each named entity and relationship recognized within a news item and for each entity and relationship within the knowledge graph.

Let $\text{similarity}(m_i, \text{kg\_mention})$ represent the cosine similarity score between a named entity $m_i$ from the news items and its counterpart in the knowledge graph, denoted as kg\_mention. This similarity score is determined based on the semantic similarity between the textual representations of the entities, computed using Word2Vec embeddings. Mathematically, the cosine similarity between two word vectors $\mathbf{v}$ and $\mathbf{w}$ is given by:

$$\text{cosine\_similarity}(\mathbf{v}, \mathbf{w}) = \frac{\mathbf{v} \cdot \mathbf{w}}{\|\mathbf{v}\| \|\mathbf{w}\|} \tag{5}$$

where $\mathbf{v} \cdot \mathbf{w}$ represents the dot product of the two vectors, and $\|\mathbf{v}\|$ and $\|\mathbf{w}\|$ represent their respective Euclidean norms. To identify the aligned KG entity for each entity mentioned $m_i$, we select the one with the highest alignment score, calculated using Equation 6:

$$\text{aligned\_entity}(m_i) = \arg\max \text{score}(m_i) \tag{6}$$

Similarly, alignment scores are computed for relationships $r_j$ and matched KG relationships are determined. This process generates sets of aligned entities $\mathcal{E}_{\text{aligned}}$ and aligned relationships $\mathcal{R}_{\text{aligned}}$. These aligned entities and relationships collectively contribute to creating a more meaningful representation of the news item, capturing the interconnections between entities. The embedding of the news item based on alignment, denoted as $\mathbf{E}_{\text{graph}}(X)$, is computed by aggregating the embeddings of aligned entities and relationships, with the alignment scores for more accurate contextualization:

$$\mathbf{E}_{\text{graph}}(X) = \frac{1}{|\mathcal{E}_{\text{aligned}}| + |\mathcal{R}_{\text{aligned}}|} \left( \sum_{e \in \mathcal{E}_{\text{aligned}}} \text{score}(e) \cdot \mathbf{v}_e + \sum_{r \in \mathcal{R}_{\text{aligned}}} \text{score}(r) \cdot \mathbf{v}_r \right) \tag{7}$$

Equation 7 represents the combination operation applied to embeddings, where $\mathbf{v}_{\text{combined}}$ denotes the combined embedding. Here, $|\mathcal{E}_{\text{aligned}}|$ and $|\mathcal{R}_{\text{aligned}}|$ represent the sizes of the sets of aligned entities and relationships, respectively. The embeddings $\mathbf{v}_e$ and $\mathbf{v}_r$ are extracted using SimplE, a model designed for capturing complex structural information in knowledge graphs. The addition symbol "+" signifies the fusion of different embeddings, achieved through a weighted sum approach. By incorporating alignment scores, aligned entities, and relationships, our fact verification system leverages factual alignment to enhance accuracy and contextual understanding.

## The integration of content and knowledge

The next crucial stage in our methodology is concatenating knowledge-based graph embeddings with the contextual embeddings obtained from DistilBERT. This combination is essential for enhancing the representation of claim embeddings. Mathematically, we represent this merging as:

$$E(X) = E_{\text{graph}}(X) + E_{\text{context}}(X) \tag{8}$$

Equation 8 signifies the integration of two essential components: the graph-based embedding capturing alignment-driven context ($E_{\text{graph}}(X)$) and the contextual embedding derived from DistilBERT's semantic understanding ($E_{\text{context}}(X)$). The combination of $E_{\text{graph}}(X)$ and $E_{\text{context}}(X)$ results in a more comprehensive representation of the news item. $E_{\text{graph}}(X)$ provides structured knowledge extracted from the knowledge graph, including relationships and entities relevant to the topic, while $E_{\text{context}}(X)$ captures the nuanced semantic understanding derived from the textual content of the news item. Integrating these two embeddings allows us to leverage the factual knowledge encoded in the knowledge graph and the contextual understanding derived from the news item's language. This super representation enables a richer interpretation of the news item, incorporating factual knowledge and contextual relevance.

## Predictive modeling

The last step of our method involves using a Multilayer Perceptron (MLP) for final prediction. MLP works well for this task because it combines different types of information. This includes insights from both the content and the graph. In mathematical terms, we can express the MLP's prediction like this:

$$\hat{y}_i = \text{MLP}\big(\text{E}(\text{X}_i)\big) \tag{9}$$

In Equation 9, $\hat{y}_i$ predicts what's true for the $i$-th news piece, and $\text{E}(\text{X}_i)$ represents the combined information we got from both the content and the graph.

We employed the MLP to classify the news items. It effectively learns from the combined information encapsulated within $E(X)$, allowing it to make informed predictions about the content's characteristics. Fig 2 offers a visual overview of our methodology. Algorithm 1 outlines various steps for the proposed cogiGraph model.

**Algorithm 1. CogiGraph**

```
 1: INPUT: Raw tweets X
 2: OUTPUT: Predictions ŷ
 3: X_processed ← PreprocessTweets(X)   {Text Preprocessing}
 4: E_context(X) ← DistilBERT(X) {Semantic Encoding}
 5: G ← BuildKG(COVID-19 ontology)   {Construct Knowledge Graph}
 6: G_enriched ← EnrichKG(G, CORD-19 dataset) {Enrich KG (see [59])}
 7: Triples ← OpenIE(X) {Extract Triples}
 8: Compute alignment scores using Equation 4
 9: E_aligned, R_aligned ← GetAlignedEntities(alignment scores)
10: E_graph(X) ← ComputeKGEmbeddings(E_aligned, R_aligned)
11: E(X) ← E_context(X) + E_graph(X) {Integrate Content and Knowledge}
12: ŷ ← MLP(E(X)) {Predict with MLP}
13: return ŷ
```

In subsequent sections, we will delve deeper into the dataset's characteristics and various implementation details of our methodology, including how the model is evaluated.

## Experimental setup

In this section, we discuss how we conducted our experiments to assess the effectiveness of the proposed CogiGraph model for fake news detection. We provide details about the dataset we used, explain how our model is set up, describe how we fine-tuned its parameters, and share the metrics we used to evaluate its performance.

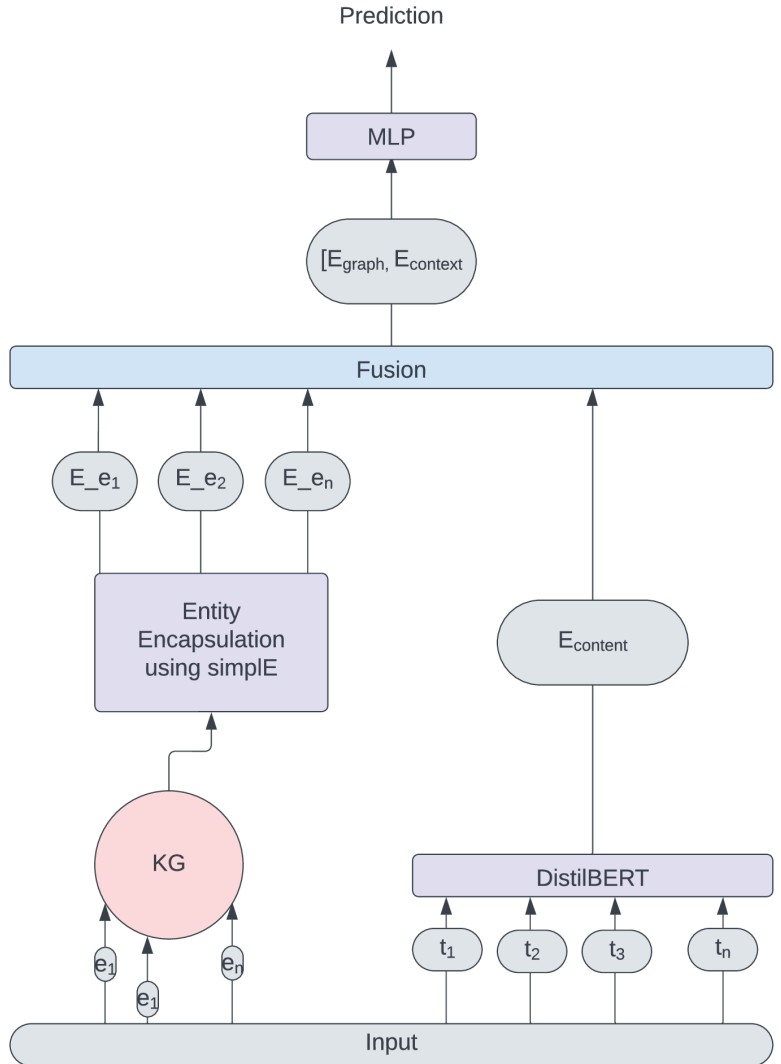

**Fig 2. Framework of cogiGraph.**

## Dataset details

First, we provide insights into the dataset utilized for evaluating our approach: the "Constraint@AAAI 2021 COVID-19" [19] fake news detection dataset. This dataset consists of 10,700 instances, including social media posts and news articles, each labeled as genuine or false. To ensure fair evaluation, the dataset has been divided into training, validation, and test sets in a balanced manner, maintaining similar proportions of genuine and false instances.

The dataset is split into train (60%), validation (20%), test (20%). The dataset is class-wise balanced as 52.34% of the samples consist of real news and 47.66% of the data consists of fake news. Moreover, we maintain the class-wise distribution across train, validation, and test splits.

Genuine news instances were sourced from various articles from reputable news outlets. False news instances were carefully curated using third-party fact-checking sources such as

NewsChecker and PolitiFact. This comprehensive dataset allows us to examine the robustness of our approach across various scenarios.

## Implementation details

Moving on to the implementation, our framework, the CogiGraph model, is designed to assess the credibility of a news item based solely on the news text and consists of two interconnected modules. The first module employs DistilBERT to generate rich embeddings from news tokens, capturing contextual meaning. The second module leverages a knowledge graph-based approach to represent entities and their relations, enhancing content understanding. These modules collectively form a combined representation that captures content and entity semantics. In this section, we provide a technical overview of the CogiGraph framework.

**Data Splitting and Balance:** The dataset has been divided into training (60%), validation(20%) and testing (20%) subsets. In this split, a balanced distribution of genuine and false instances has been maintained across all sets to ensure unbiased evaluation.

**Hyperparameter configuration:** We used distinct hyperparameters for various components, including DistilBERT, Bio-BERT, Knowledge Graph Enrichment, and MLP Parameters. These hyperparameters were selected based on experimentation to balance computational efficiency and convergence speed.

- *DistilBERT*: For semantic encoding using DistilBERT, we employed a batch size of 64 and a learning rate of $2 \times 10^{-5}$, determined through experimentation to balance computational efficiency and convergence speed. The model was optimized for a sequence length of 128 tokens, and the fine-tuning process consisted of training for five epochs.
- *BioBERT*: Within the implementation framework, the Bio-BERT-based Named Entity Recognition (NER) model plays an important role in accurately identifying named entities within news text. The model utilizes the BioBERT model, loaded with pre-trained weights and fine-tuned on a pre-processed version of the CORD-19 dataset specifically curated for recognizing named entities within the COVID-19 domain. We use a dropout layer for regularization and a linear output layer with softmax activation to predict entity labels. Regarding training parameters, we use an AdamW optimizer with a batch size of 64 and a learning rate of $5 \times 10^{-5}$, trained for 30 epochs. The model's training employs the cross-entropy loss function, optimizing the model through back-propagation while preventing gradient explosion using gradient clipping. Integration of the BioBERT NER model within our framework enhances entity identification.
- *Knowledge graph enrichment:* Utilizing the fine-tuned BioBERT model for entity recognition and the factual knowledge from a cleaned subset of the CORD-19 dataset consisting of abstracts and introductions of recent scientific research resulted in a rich and structured knowledge graph.
- *MLP parameters:* The Multilayer Perceptron (MLP), our classification head, comprises two hidden layers comprising 128 units each and utilizing a ReLU activation function. The final classification layer has a softmax activation function for binary classification. We opted for a learning rate of 0.0001 for effective optimization.

Fig 3 demonstrates the effectiveness of the proposed model, as the graph precisely illustrates the performance of both training and validation accuracy. Significantly, a plateau is attained after 150 epochs, highlighting the model's stability and resilience. During the experiment, a careful incorporation of K-fold cross-validation was used, with a predetermined value of K set at 10, to guarantee a thorough assessment of the model's performance.

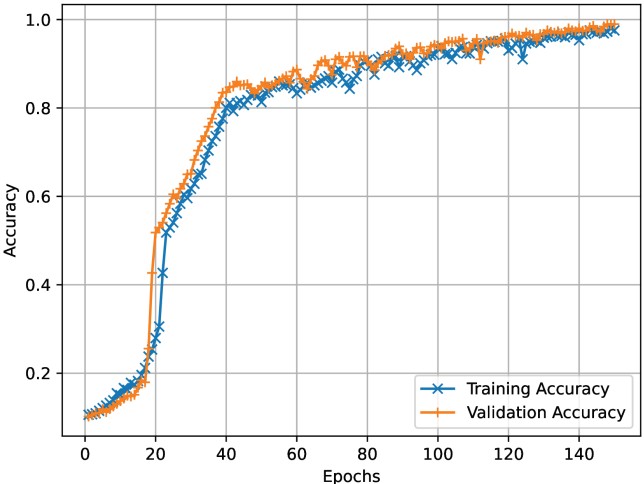

**Fig 3. Accuracy plot of CogiGraph model.**

Fig 4 visually represents the complex interplay between training and validation loss across several epochs, offering a detailed comprehension of the model's learning trajectory. The distinguishing factor of the suggested technique lies in its notable attainment of a 98.97% accuracy, accompanied by a strong sensitivity of 99.01% and a formidable F1 score of 98.92. The measurements demonstrate the higher performance of the proposed model, surpassing other state-of-the-art methodologies. Table 1 is a comprehensive compilation of findings, thoroughly juxtaposing the proposed technique and many other innovative methods. The proposed approach was thoroughly assessed using several metrics, such as precision, recall, and the F1 score. This evaluation proved its efficacy and dependability more than current methodologies.

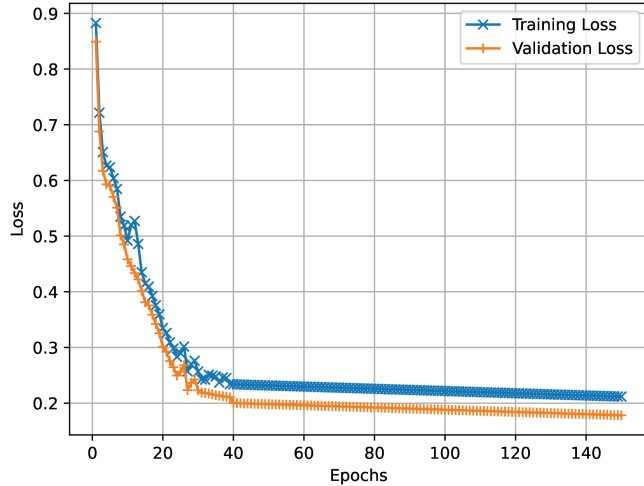

**Fig 4. Loss plot of CogiGraph model.**

**Table 1. Comparative study with other state-of-the-art methods.**

| Model | Accuracy | Precision% | Recall% | F1 Score |
|---|---|---|---|---|
| BERT-BiGRU-Attention-CapsNET-(BiGRU-CRF) [61] | 0.9196 | - | 0.9193 | 0.9113 |
| CNN + BiLSTM [62] | 0.9201 | 0.9201 | 0.9201 | 0.9201 |
| GRAPHSAGE [63] | - | - | - | 0.9280 |
| SVM [19] | 0.9332 | 0.9333 | 0.9332 | 0.9332 |
| SVM + TF-idf [62] | 0.9435 | 0.9442 | 0.9439 | 0.9439 |
| LR + ULMFit + BERT [64] | 0.9700 | - | - | 0.9800 |
| CT-BERT+BiGRU [65] | 0.9846 | 0.9797 | 0.9911 | 0.9854 |
| Text-Transformers + Five-fold five-model cross-validation + Pseudo Label Algorithm [66] | 0.9850 | 0.9860 | 0.9850 | 0.9850 |
| Roberta-base [67] | 0.9864 | 0.9865 | 0.9864 | 0.9864 |
| CT-BERT Ensemble [68,69] | 0.9869 | 0.9869 | 0.9869 | 0.9869 |
| Ensemble voting classifier (LR, CART, LSVM) [70] | 0.9879 | 0.9888 | 0.9888 | 0.9888 |
| **SFFN + Heuristic Post-Processing** [71] | 0.9892 | **0.9892** | 0.9892 | **0.9892** |
| **CogiGraph** | **0.9897** | 0.9882 | **0.9901** | **0.9892** |

**Model training and fusion:** Our approach's effectiveness mostly relies on finetuning, extracting named entities, populating the knowledge graph and the training process, especially given the size of the models and the size of the CORD-19 dataset. To address this, we harnessed the computational power of the NVIDIA AI Server (DGX A100). We used model parallelism, which involves partitioning a neural network model across multiple GPUs for optimized distributed training. Model parallelism allows us to distribute different sub-networks of the model across the available GPUs, thereby overcoming memory limitations and splitting the BioBERT model across multiple available GPUs as its size surpasses the individual memory available to a single GPU.

1. **Semantic encoding and named entity recognition:** Through DistilBERT, we extracted contextual embeddings $E_{\text{context}}(X)$ from the news content, as shown in equation 1. The BioBERT-based NER model, fine-tuned with the selected hyperparameters, accurately identified COVID-19-relevant named entities. The graph embeddings for the news item X are calculated using equation 7.
2. **Alignment and fusion:** The calculated alignment score and the aligned entities and relationships help us align the claim entities with those of KG entities to get the most proximate ones. The final graph embedding is calculated using the alignment scores and the embeddings of the aligned entities and relationships from the KG. The alignment score and the aligned entities calculation are shown in equations 4 and 6, respectively. The fusion step combines $E_{\text{context}}(X)$ and KG-based embeddings $E_{\text{graph}}(X)$ into a unified representation $E(X)$, as shown in equation 8.
3. **Predictive modeling:** Our MLP classification head, trained with the chosen hyperparameters, leverages $E(X)$ to predict the authenticity of news items. Using the Adam optimizer with the designated learning rate, combined with the mixture of content and knowledge-driven insights, enables accurate predictions.

The hyperparameters at various stages were selected experimentally to increase the efficiency of the CogiGraph framework. The use of linguistic analysis, together with entity recognition and the knowledge graph, helped us construct a robust system to verify news items from just the news text.

## Results and discussion

In this section, we present the results of our experiments and provide an in-depth analysis of the performance of the CogiGraph model. We also compare its performance against other state-of-the-art approaches to assess its efficacy.

### Performance comparison

For performance comparison, selecting appropriate metrics is crucial. The metrics are chosen based on their ability to capture various aspects of classification performance, such as accuracy, precision, recall, and F1-score. These metrics ensure a comprehensive evaluation of the model's ability to distinguish between fake and legitimate news.

For instance, accuracy measures the overall correctness of predictions but may not reflect the performance in imbalanced datasets. Precision and recall, on the other hand, are particularly important for understanding the model's effectiveness in identifying fake news without false alarms. The F1-score provides a harmonic mean of precision and recall, offering a balanced view when trade-offs exist between these metrics.

In addition, recent studies emphasize the importance of advanced evaluation techniques for fake news detection. For example, one work highlights the need for robust metric selection tailored to specific challenges in misinformation detection, including imbalanced data distributions and varying contexts [72]. Moreover, it is critical to ensure that the metrics align with the objectives of the detection system, whether the focus is on minimizing false negatives (critical in sensitive scenarios) or maximizing true positives (important for broad misinformation campaigns).

Table 1 summarizes the performance of the CogiGraph model and selected state-of-the-art approaches on the "Constraint@AAAI 2021 COVID-19" dataset. The metrics used for evaluation include accuracy, precision, recall, and F1-score because of the balanced nature of the dataset and for better comparison insights.

- *Karnyoto et al. [61]*: This model comprises a Bidirectional GRU layer receiving input from BERT, followed by an Attention Layer for feature extraction, a Capsule Network Layer for nuanced neuron connections, and a BiGRU-CRF layer for segmentation and processing.
- *Sharif et al. [62]*: A combination of CNN and BiLSTM was used here. In the CNN component, 64 convolution filters with a kernel size 5x5 are applied, along with a pooling window size of 1x5. The BiLSTM network consists of 32 bidirectional cells with a dropout rate 0.2. These two networks are then sequentially integrated into the combined model.
- *Karnyoto et al. [63]*: This model incorporates word co-occurrence and TF-IDF to establish edges in three graph models: Graph Convolutional Network, Graph Attention Network, and GraphSAGE. Augmentation techniques like random deletion, insertion, swap, and synonym replacement were applied
- *Patwa et al. [19]*: This model highlights SVM-based classification as the top-performing method among various machine learning models.
- *Sharif et al. [62]*: This model leverages a combination of SVM and TF-IDF. The SVM component utilizes TF-IDF features for classification, achieving notable performance.
- *Biradar et al. [64]*: This model combines a language model with conventional machine learning algorithms using a voting classifier approach, achieving the highest accuracy with an ensemble setting of LR, ULMFit classifier, and BERT classifier.
- *Alghamdi et al. [65]*: This model leveraged Bidirectional Gated Recurrent Units(BiGRU) over pre-trained CT-BERT architecture.

- *Li et al. [66]*: The authors utilized a transformer-based architecture and incorporated five-fold five-model cross-validation and the pseudo label algorithm. The findings suggest that the ensemble approach and integrating cross-validation and pseudo-labeling strategies contribute to enhanced performance metrics.
- *Raha et al. [67]*: The authors went for RoBERTa-base: 12-layer, 768-hidden, 12-heads, 125M parameters. Trained on a larger dataset with increased iterations and a batch size of 8k, RoBERTa removes the NSP objective during pretraining.
- *Patwa et al. [68]*: The authors have proposed a simple but effective approach to COVID-19 fake news detection, utilizing CT-BERT and ensemble learning techniques. Their experiments validate the effectiveness of BERT-based models in subject-specific tasks, achieving high-quality binary classification.
- *Varshney et al. [70]*: The authors Employed an Ensemble-based model incorporating Logistic Regression (LR), Linear Support Vector Machine (LSVM), and Classification and Regression Trees (CART), where their voting gives the final decision.
- *Das et al. [71]*: This method combines metadata with an ensemble of pre-trained language models for fake news classification. It includes Text Preprocessing, Tokenization, Backbone Model Architectures, Ensemble, Statistical Feature Fusion Network, Predictive Uncertainty Estimation Model, and Heuristic Post-Processing.

Our approach outperformed the competition's leaderboard regarding all the evaluation metrics for this shared task, where this dataset was released initially. Compared with the Statistical Feature Fusion Network with MCDropout (SFFN) and Post-Processing approach [71], our approach performed better in accuracy and recall and achieved a similar F1 score. SFFN is a statistical feature fusion network that incorporates the dropout technique known as Monte Carlo Dropout (MCDropout). This method helps to mitigate overfitting and improve the model's generalization performance. In the revised manuscript, we will provide a detailed explanation of SFFN and its role in our proposed approach to ensure clarity for readers.

## Conclusion

The proposed CogiGraph framework successfully addresses the challenge of fake news detection by integrating content-based features with knowledge graph insights. By leveraging advanced NLP models such as DistilBERT and BioBERT, and enhancing representation through SimplE embeddings, the framework offers a holistic solution to misinformation detection in the biomedical domain. Experimental results demonstrate that CogiGraph outperforms state-of-the-art methods across multiple evaluation metrics, achieving high accuracy and F1 scores.

This research makes several notable contributions:

- *Unified framework*: The seamless fusion of semantic content analysis and entity-level evidence-based reasoning sets a new benchmark for misinformation detection systems.
- *Domain-Specific Effectiveness*: By adopting domain-specific biomedical models and knowledge graphs, the system achieves higher accuracy in health-related misinformation scenarios.
- *Scalability and Generalizability*: The framework is adaptable to various domains beyond COVID-19, providing a strong foundation for future misinformation detection systems.

### Limitations

Despite its promising outcomes, the CogiGraph framework has certain limitations:

- *Multimodal content handling*: The current system primarily focuses on textual data and lacks the capability to process multimedia content, such as images and videos, which are increasingly prevalent in misinformation.
- *Computational complexity*: Maintaining and updating large-scale knowledge graphs can be computationally intensive and may require significant resources for real-time processing.
- *Dynamic knowledge integration*: The static nature of the current knowledge graph structure limits its adaptability to real-time updates and evolving information landscapes.
- *Domain dependency*: The reliance on biomedical-specific models and datasets may reduce the system's performance in other domains without additional fine-tuning and data preparation.
- *Dataset generalizability*: Our experiments have been conducted on a single dataset, which may limit the generalizability of the findings. Future work should incorporate additional datasets to further validate the framework across diverse scenarios.
- *Explainability:* The current framework lacks sophisticated methods for providing transparent and interpretable detection reasoning, which is crucial for practical adoption.

In conclusion, the CogiGraph framework represents a significant advancement in combating misinformation by unifying content and conducting evidence-based analyses. As digital communication continues to evolve, this research lays a foundation for developing sophisticated tools that safeguard information integrity and promote informed public discourse.

## Future scope

In the future, CogiGraph holds great promise for tackling fake news. We envision continuously updating its knowledge base with real-time information and incorporating other modalities like images and videos for a richer understanding. To make its reasoning process transparent, we plan to develop methods for explaining predictions and highlighting key information. Additionally, adapting CogiGraph to diverse domains and evaluating it on larger, multi-lingual datasets will enhance its generalizability and fairness. By exploring these avenues, we aim to refine CogiGraph and develop robust tools for combating fake news across various domains and languages, empowering individuals to discern truth from fiction in our information-rich world.

## Abbreviations

MLP: Multilayer perceptron
NER: Named entity recognition
Kg: Knowledge hraph
COVID-19: Coronavirus Disease 2019
CORD-19: COVID-19 Open Research Dataset
SVM: Support vector machine
TF-idf: Term frequency-inverse document frequency
LR: Logistic regression
F1: F1 score

## Author contributions

**Conceptualization:** Rayees Ahmad Dar, Rana Hashmy.

**Data curation:** Rayees Ahmad Dar.

**Formal analysis:** Rayees Ahmad Dar.

**Funding acquisition:** Rayees Ahmad Dar, Muhammad Shahid Anwar.

**Investigation:** Rayees Ahmad Dar.

**Methodology:** Rayees Ahmad Dar.

**Project administration:** Rayees Ahmad Dar.

**Resources:** Rayees Ahmad Dar.

**Software:** Rayees Ahmad Dar.

**Supervision:** Rana Hashmy, Muhammad Shahid Anwar, Patrik Böhm, Jaroslav Frnda.

**Validation:** Rayees Ahmad Dar.

**Visualization:** Rayees Ahmad Dar.

**Writing – original draft:** Rayees Ahmad Dar.

**Writing – review & editing:** Rayees Ahmad Dar.

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
