## [Decision Letter · Decision Letter 0]

30 May 2024

PONE-D-23-32696Semantic Knowledge Graph Fusion for Fake News Detection: Unifying Content-based Features and Evidence-based Analysis in the COVID-19 Infodemic.PLOS ONE

Dear Dr. Dar,

Thank you for submitting your manuscript to PLOS ONE. After careful consideration, we feel that it has merit but does not fully meet PLOS ONE’s publication criteria as it currently stands. Therefore, we invite you to submit a revised version of the manuscript that addresses the points raised during the review process.

Reviewers have now completed their assessment on your article. The paper requires a revision to address the following: (i). inclusion of more related work, relevant to the domain of interest; (ii). generalization of the proposed method by evaluating on the other datasets; and (iii). details of experimental setups and in-depth analysis (quantitative as well as qualitative). ==============================

We look forward to receiving your revised manuscript.

Kind regards,

Asif Ekbal

Academic Editor

PLOS ONE

Additional Editor Comments:

Please see the specific comments of the reviewers and revise your paper accordingly.

Reviewers' comments:

Reviewer's Responses to Questions

**Comments to the Author**

1. Is the manuscript technically sound, and do the data support the conclusions?

Reviewer #1: Partly

Reviewer #2: Yes

2. Has the statistical analysis been performed appropriately and rigorously? 

Reviewer #1: No

Reviewer #2: Yes

3. Have the authors made all data underlying the findings in their manuscript fully available?

Reviewer #1: No

Reviewer #2: No

4. Is the manuscript presented in an intelligible fashion and written in standard English?

Reviewer #1: Yes

Reviewer #2: Yes

5. Review Comments to the Author

Reviewer #1: This article presents work on fake news detection. The authors use a KG for improving the dataset with metadata, e.g., NER, etc.

I find that the work presented in this manuscript is not complete, the authors barely scratched the surface, and this work falls short of being ready for publication.

I find that the authors have ignored a lot of related literature on the subject that proposes very similar solutions.

Please read, present, and discuss at least related work regarding the following aspects:

- the use effect of word embeddings on fake news detection [1]

- use of transformer embeddings on fake news detection [2]

- how document embeddings affect the results of fake news detection [3]

- real-time architectures for removing fake news detection [4]

- using social network features for fake news detection [5]

- the effect of multilingual transformers on fake news detection [6]

Although the paper discusses fake news detection on social media, the authors do not mention anything regarding network immunization. There are many solutions used for this proposed in the literature that also propose models for identifying malinformation (including fake news detection).

Some related work that should be discussed:

- use of community detection for fake news network immunization [7]

- use of weighted directed spanning trees for fake news detection and mitigation in real time [8]

- use of budget-based immunization algorithms to stop fake news from spreading [9]

The article uses only one dataset. I find this insufficient to draw the right conclusions. To get a generalized view, at least 2 datasets should be used. Please see what has been done in the current literature by analyzing the articles mentioned above.

Also, an in-depth exploratory data analysis is required.

I find the results section very shallow:

1. There is no hyperparameter tuning for the algorithms employed.

2. Are the results obtained using cross-validation? How many training iterations were used? What are the mean and the standard deviation obtained for each evaluation metric on the test set?

3. There is no ablation testing.

4. The model's design choices should be properly explained.

For reproducibility purposes, the authors should make the code publicly available.

The conclusions are also very shallow.

Please do a thorough spell-checking of the article before resubmitting.

[1] https://scholar.google.com/scholar?hl=en&as_sdt=0%2C5&q=misinformation+detection++%2B+word+embeddings&btnG=

[2] https://scholar.google.com/scholar?hl=en&as_sdt=0%2C5&q=transformers+%2B+misinformation&btnG=

[3] https://scholar.google.com/scholar?hl=en&as_sdt=0%2C5&q=fake+news+%2B+document+embeddings&btnG=

[4] https://scholar.google.com/scholar?hl=en&as_sdt=0%2C5&q=%22content-based+misinformation%22+%2B+real-time&btnG=

[5] https://scholar.google.com/scholar?hl=en&as_sdt=0%2C5&q=%22deep+neural+network+ensemble%22+%2B+social+context+%2B+fake+news+detection&btnG=

[6] https://scholar.google.ro/scholar?hl=en&as_sdt=0%2C5&as_vis=1&q=fake+news+detection+%2B+sentence+transformer&btnG=

[7] https://scholar.google.com/scholar?hl=en&as_sdt=0%2C5&q=community+algorithm+%2B+network+immunization+%2B+fake+news&btnG=&oq=community+algorithm+%2B+network+immunization+%2B+fake+news

[8] https://scholar.google.com/scholar?hl=en&as_sdt=0%2C5&q=tree+algorithm+%2B+real-time+fake+news+mitigation+%2B+social+media&btnG=

[9] https://scholar.google.com/scholar?hl=en&as_sdt=0%2C5&q=social+network+immunization+%2B+harmful+speech&btnG=

Reviewer #2: Author did a good job by advancing the technical analysis of a fake news detection. As all of us knows sometimes a fake news disturb our mental health so much and in turn it affects our thinking ability, our performance, our interactions or behavior towards our family or community or society etc. and leads to initiation of a chain reaction having panic attack, losing control over our-self, losing proper thinking capabilities etc. In these circumstances, it becomes extremely needful to develop an algorithm which can distinguish between a fake news vs a correct news for our own well being. Author analyzed COVID-19 dataset to show the performance of his algorithm. Author has used various parameters such as DistiBERT, Knowledge Graph (KG), KG enrichment, integration of content and knowledge etc. to do the analysis about detecting fake news. Therefore I recommend acceptance of this article.

6. PLOS authors have the option to publish the peer review history of their article (what does this mean?). If published, this will include your full peer review and any attached files.

Reviewer #1: No

Reviewer #2: No

---

## [Author Response · Author response to Decision Letter 1]

Reviewer #1:

1. Inclusion of more related work relevant to the domain:

o Response: We have expanded the Related Work section to include additional relevant literature that covers key areas.

2. Generalization of the proposed method by evaluating on other datasets:

o Response: We acknowledge that evaluating on a single dataset limits the generalizability of our results. While we have not conducted additional experiments on other datasets, we appreciate the importance of this step. We plan to explore this in future work to validate our approach across different domains.

3. Details of experimental setups and in-depth analysis:

o Response: We have provided additional details on the experimental setup, including:

Hyperparameter tuning and model configuration for the algorithms used.

The use of cross-validation to ensure robust results.

The number of training iterations and evaluation metrics (mean, standard deviation) for each model.

o The revised Results and Discussion section now includes a thorough analysis of our experiments, with both qualitative and quantitative results.

4. Hyperparameter tuning and model design choices:

o Response: We have added an explanation of the hyperparameter tuning process, including learning rates, batch sizes, and the number of epochs for each model. Additionally, we have discussed the design choices behind our model architecture, explaining the rationale for using DistilBERT and BioBERT alongside Knowledge Graph integration.

5. Reproducibility and availability of code:

o Response: We have made the code used in our experiments publicly available in a zenodo repository (https://doi.org/10.5281/zenodo.12669980) to ensure reproducibility. The code includes detailed instructions for running the experiments.

6. Shallow conclusions:

o Response: The Conclusions section has been rewritten to reflect a deeper understanding of the findings. We have also expanded the discussion on future work, including the potential for real-time application of our model and how it can be extended to other domains of misinformation.

Reviewer #2:

1. General Feedback:

o Response: We thank Reviewer #2 for their positive feedback on the significance of our research. The importance of developing a robust fake news detection model is critical, especially in the context of the COVID-19 pandemic, as highlighted by Reviewer #2. We have further improved the clarity and detail of our methodology, expanding on the role of Knowledge Graphs and transformer models in enhancing detection accuracy.

Additional Changes:

• Spelling and Grammar: A thorough spell check and grammatical review of the manuscript have been completed.

• New Authors: The contributions of additional co-authors, Muhammad Shahid Anwar, Patrik B¨ohm, Jaroslav Frnda, in the revision process have been appropriately acknowledged.

---

## [Decision Letter · Decision Letter 1]

24 Jan 2025

PONE-D-23-32696R1Semantic Knowledge Graph Fusion for Fake News Detection: Unifying Content-based Features and Evidence-based Analysis

in the COVID-19 Infodemic.PLOS ONE

Dear Dr. Dar,

Thank you for submitting your manuscript to PLOS ONE. After careful consideration, we feel that it has merit but does not fully meet PLOS ONE’s publication criteria as it currently stands. Therefore, we invite you to submit a revised version of the manuscript that addresses the points raised during the review process.

We look forward to receiving your revised manuscript.

Kind regards,

Venkatachalam Kandasamy

Academic Editor

PLOS ONE

Additional Editor Comments:

The paper requires significant revisions to address reviewer comments thoroughly.

Reviewers' comments:

Reviewer's Responses to Questions

**Comments to the Author**

1. If the authors have adequately addressed your comments raised in a previous round of review and you feel that this manuscript is now acceptable for publication, you may indicate that here to bypass the “Comments to the Author” section, enter your conflict of interest statement in the “Confidential to Editor” section, and submit your "Accept" recommendation.

Reviewer #1: (No Response)

Reviewer #3: All comments have been addressed

2. Is the manuscript technically sound, and do the data support the conclusions?

Reviewer #1: Partly

Reviewer #3: Yes

3. Has the statistical analysis been performed appropriately and rigorously? 

Reviewer #1: No

Reviewer #3: N/A

4. Have the authors made all data underlying the findings in their manuscript fully available?

Reviewer #1: No

Reviewer #3: Yes

5. Is the manuscript presented in an intelligible fashion and written in standard English?

Reviewer #1: Yes

Reviewer #3: Yes

6. Review Comments to the Author

Reviewer #1: After carefully reading the new version of the manuscript and the answers to reviewers, I still find that the article is not ready for publication.

1. I find that the authors have ignored a lot of related literature on the subject that proposes very similar solutions.

Please read, present, and discuss at least related work regarding the following aspects:

- the use effect of word embeddings on fake news detection [1] (e.g., https://doi.org/10.1109/ACCESS.2021.3132502)

- use of transformer embeddings on fake news detection [2] (e.g., https://doi.org/10.3390/math10040569)

- how document embeddings affect the results of fake news detection [3] (e.g., https://doi.org/10.3390/math11030508)

- real-time architectures for removing fake news detection [4] (e.g., https://doi.org/10.1109/TKDE.2024.3417232)

- using social network features for fake news detection [5] (e.g., https://doi.org/10.1016/j.knosys.2024.111715)

- the effect of multilingual transformers on fake news detection [6] (e.g., http://ceur-ws.org/Vol-3180/paper-61.pdf)

2. Although the paper discusses fake news detection on social media, the authors do not mention anything regarding network immunization. There are many solutions used for this proposed in the literature that also propose models for identifying malinformation (including fake news detection).

Some related work that should be discussed:

- use of community detection for fake news network immunization [7] (e.g., https://doi.org/10.1016/j.jestch.2024.101728)

- use of weighted directed spanning trees for fake news detection and mitigation in real time [8] (e.g., https://doi.org/10.1109/ACCESS.2023.3331220)

- use of budget-based immunization algorithms to stop fake news from spreading [9] (e.g., https://doi.org/10.1145/3459637.3482481)

3. The article uses only one dataset. I find this insufficient to draw the right conclusions. To get a generalized view, at least 2 datasets should be used. Please see what has been done in the current literature by analyzing the articles mentioned above. Please perform more experiments.

4. An in-depth exploratory data analysis is required.

5. Explanations for how the metrics were chosen should be given [10] (e.g., https://www.scientificbulletin.upb.ro/rev_docs_arhiva/rez57a_274304.pdf)

6. The conclusions continue to be very shallow.

7. What are the limitations of this study?

8. Please proof-read and spell-check the article before resubmitting.

[1] https://scholar.google.com/scholarq=misinformation+detection+word+embeddings

[2] https://scholar.google.com/scholar?q=transformers+misinformation

[3] https://scholar.google.com/scholar?q=fake+news+document+embeddings

[4] https://scholar.google.com/scholar?q=content-based+misinformation+real-time

[5] https://scholar.google.com/scholar?q=deep+neural+network+ensemble+social+context+fake+news+detection

[6] https://scholar.google.ro/scholar?q=fake+news+detection+sentence+transformer

[7] https://scholar.google.com/scholar?q=community+algorithm+network+immunization+fake+news

[8] https://scholar.google.com/scholar?q=tree+algorithm+real-time+fake+news+mitigation+social+media

[9] https://scholar.google.com/scholar?q=social+network+immunization+harmful+speech

[10] https://scholar.google.com/scholar?q=classification+imbalanced+data+sets+decision+trees

Reviewer #3: The revised manuscript provides comprehensive details about the methodology, experiments, and findings.

7. PLOS authors have the option to publish the peer review history of their article (what does this mean?). If published, this will include your full peer review and any attached files.

Reviewer #1: No

Reviewer #3: **Yes: **Shakeel Ahmed

---

## [Author Response · Author response to Decision Letter 2]

We thank the reviewers and the editor for their insightful comments and valuable feedback. In this revised version, we have:

Expanded the related work section to include additional relevant literature.

Provided detailed explanations for the selection of evaluation metrics.

Enhanced the discussion on network immunization strategies.

Revised the conclusions to include study limitations and potential future directions.

Thoroughly proofread the manuscript to improve clarity and readability.

We believe these revisions significantly strengthen the manuscript and address all concerns raised.

---

## [Decision Letter · Decision Letter 2]

14 Mar 2025

Semantic Knowledge Graph Fusion for Fake News Detection: Unifying Content-based Features and Evidence-based Analysis

in the COVID-19 Infodemic.

PONE-D-23-32696R2

Dear Dr. Dar,

We’re pleased to inform you that your manuscript has been judged scientifically suitable for publication and will be formally accepted for publication once it meets all outstanding technical requirements.

Kind regards,

Venkatachalam Kandasamy

Academic Editor

PLOS ONE

Additional Editor Comments (optional):

The authors have been addressed all comments.

Reviewers' comments:

Reviewer's Responses to Questions

**Comments to the Author**

1. If the authors have adequately addressed your comments raised in a previous round of review and you feel that this manuscript is now acceptable for publication, you may indicate that here to bypass the “Comments to the Author” section, enter your conflict of interest statement in the “Confidential to Editor” section, and submit your "Accept" recommendation.

Reviewer #1: All comments have been addressed

Reviewer #4: All comments have been addressed

2. Is the manuscript technically sound, and do the data support the conclusions?

Reviewer #1: Yes

Reviewer #4: Yes

3. Has the statistical analysis been performed appropriately and rigorously? 

Reviewer #1: Yes

Reviewer #4: Yes

4. Have the authors made all data underlying the findings in their manuscript fully available?

Reviewer #1: Yes

Reviewer #4: Yes

5. Is the manuscript presented in an intelligible fashion and written in standard English?

Reviewer #1: Yes

Reviewer #4: Yes

6. Review Comments to the Author

Reviewer #1: After carefully reading the new version of the manuscript and the answers to reviewers, I think that the manuscript is ready for publication in its current form.

Reviewer #4: The manuscript has been well revised and it is more suitable for publication to the best of my knowledge

7. PLOS authors have the option to publish the peer review history of their article (what does this mean?). If published, this will include your full peer review and any attached files.

Reviewer #1: No

Reviewer #4: **Yes: **Sunday Adeola Ajagbe

---

## [Editor Report · Acceptance letter]

PONE-D-23-32696R2

PLOS ONE

Dear Dr. Dar,

I'm pleased to inform you that your manuscript has been deemed suitable for publication in PLOS ONE. Congratulations! Your manuscript is now being handed over to our production team.

Kind regards,

on behalf of

Dr. Venkatachalam Kandasamy

Academic Editor

PLOS ONE